# Which Actigraphy Dimensions Predict Longitudinal Outcomes in Bipolar Disorders?

**DOI:** 10.3390/jcm11082204

**Published:** 2022-04-14

**Authors:** Lisa Ferrand, Vincent Hennion, Ophelia Godin, Frank Bellivier, Jan Scott, Bruno Etain

**Affiliations:** 1Optimisation Thérapeutique en Neuropsychopharmacologie, INSERM U1144, Université de Paris, 75006 Paris, France; lisa.ferrand@aphp.fr (L.F.); vincent.hennion@aphp.fr (V.H.); frank.bellivier@inserm.fr (F.B.); 2AP-HP.Centre, DMU Psychiatrie et Addictologie, Université de Paris, 75006 Paris, France; 3Université de Paris, 75006 Paris, France; jan.scott@newcastle.ac.uk; 4AP-HP.Nord, GH Saint-Louis-Lariboisière-F, Widal, DMU Neurosciences, Département de Psychiatrie et de Médecine Addictologique, 75010 Paris, France; 5INSERM, IMRB, Translational Neuropsychiatry, Fondation FondaMental, 94000 Créteil, France; ophelia.godin@fondation-fondamental.org; 6Translational and Clinical Research Institute, Newcastle University, Newcastle upon Tyne NE1 7RU, UK

**Keywords:** bipolar disorder, recurrence, actigraphy, predictors, circadian rhythms, sleep, longitudinal, survival analysis

## Abstract

Bipolar disorder (BD) is characterized by recurrent mood episodes. It is increasingly suggested that disturbances in sleep–wake cycles and/or circadian rhythms could represent valuable predictors of recurrence, but few studies have addressed this question. Euthymic individuals with BD (n = 69) undertook 3 weeks of actigraphy recording and were then followed up for a median duration of 3.5 years. Principal component analyses were used to identify core dimensions of sleep quantity/variability and circadian rhythmicity. Associations between clinical variables and actigraphy dimensions and time to first recurrence were explored using survival analyses, and then using area under the curve (AUC) analyses (early vs. late recurrence). Most participants (64%) experienced a recurrence during follow-up (median survival time: 18 months). After adjusting for potential confounding factors, an actigraphy dimension comprising amplitude and variability/stability of circadian rhythms was a significant predictor of time to recurrence (*p* = 0.009). The AUC for correct classification of early vs. late recurrence subgroups was only 0.64 for clinical predictors, but combining these variables with objectively measured intra-day variability improved the AUC to 0.82 (*p* = 0.04). Actigraphy estimates of circadian rhythms, particularly variability/stability and amplitude, may represent valid predictive markers of future BD recurrences and could be putative targets for future psychosocial interventions.

## 1. Introduction

Bipolar disorder (BD) is typically characterized by a lifetime course of recurrent mood episodes, which contributes substantially to its global burden. After the first onset of BD, the estimated rates of relapse (i.e., return of an index episode) and recurrence (i.e., a new episode onset) are about 39% and 52% per annum, respectively [1]. Once individuals experience their first recurrence, they are at long-term risk of multiple episodes [2,3]. Given these findings, an important goal of BD research is to identify robust and clinically meaningful predictors of new episode onsets, especially early recurrences. 

In the last two decades, several reviews and large-scale independent studies have examined clinical predictors of recurrence (e.g., [4,5,6,7,8,9,10]). It is suggested that various socio-demographic and illness factors are associated with BD recurrence, including age; sex; childhood maltreatment; recent life events; number of prior mood episodes (and/or rapid cycling); residual depressive or other inter-episode symptoms; BD subtype; comorbid physical (e.g., thyroid disorders) or psychiatric disorders (e.g., anxiety disorders); nicotine, alcohol and/or substance misuse; overweight or obesity; and number, class of psychotropic medications and/or adherence to medications. However, findings lack consistency and, as noted by Treuer and Tohen [9], forecasting the course of BD is challenging and largely relies on clinical characteristics that together make only a modest contribution to the prediction of recurrences, with low individual predictive values [5]. Therefore, there is a long-term ambition to introduce laboratory tests or other forms of individual screening for physiological or biological markers of the course and outcome of BD [11]. Currently, the putative markers identified through applied research have not yet been transferred from bench to bedside, often because the results are not replicated and/or the techniques are experimental or are too resource or cost-intensive to be used in day-to-day practice [12]. 

Interestingly, one promising area of investigation focuses on the sleep–wake cycle and circadian rhythm disturbances. These disruptions are of interest for several key reasons. First, abnormalities in rest–activity rhythms (i.e., the combination of sleep–wake cycle, motor activity and circadian rhythm disturbances) are core symptoms of BD and exacerbations of these abnormalities are associated with BD recurrences [13,14]. Second, chronobiological models offer a plausible explanation of the observed phenomenology of BD [15]. Third, several self- and observer-rated tools can be used to evaluate sleep quality, chronotypes or other aspects of sleep–wake patterns in clinical settings [13,16]. Fourth, studies have shown that these self- or observer-rated sleep disturbances assessed during inter-episode periods are associated with BD relapse or recurrence [17,18,19,20,21]. Nevertheless, there are some limitations in published studies. For example, studies employing self and observer ratings tend to focus only on sleep quality or duration rather than chronobiological proxies, and very few projects offer reliable or valid self-assessment of other elements of rest–activity rhythms. Furthermore, some studies blur the boundaries between euthymia and residual symptoms and/or between BD relapses and recurrences, and most studies have not reported time-to-event (survival) analyses and/or determined the proportion of individuals with early or late recurrence. Nonetheless, the findings indicate that further exploration is warranted, especially using objective monitoring of sleep and circadian rhythms, such as the relatively low-cost option of wrist-worn actigraphy devices [22]. 

Until recently, existing studies of actigraphy in BD were small scale and/or they did not examine factors that may confound the associations between actigraphy and the outcomes (e.g., depressive symptom severity, body mass index, sleep apnea, medications). Critically, nearly all this research has been cross-sectional and there is limited data about longitudinal outcomes of BD after undertaking baseline actigraphy recordings [22]. However, in 2021, Esaki and colleagues reported findings from a sample of 189 participants who undertook 7 consecutive days of actigraphy and were followed up for 12 months [23]. The main findings were that a higher amplitude of circadian rhythms was significantly associated with a decrease in mood episode relapses. However, as mentioned by the authors, there are potential limitations that prevent the generalizability of the findings. For example, the sample primarily comprised individuals with BD-II (64%) and most of them (59%) reported persistent residual mood symptoms at baseline. Furthermore, the 7-day duration of actigraphy recording was probably suboptimal since guidelines and meta-analyses suggest 14–21 days is preferable, especially when assessing variability in rhythms [24,25]. Finally, mood relapses were not recorded according to recognized diagnostic criteria. Given these study limitations, it is timely to try to confirm the reported findings in an independent sample of individuals with BD-I and II, who were euthymic at baseline. Additionally, a new study offers the opportunity to use a more extended duration of actigraphy and to assess BD recurrences with valid DSM criteria. Moreover, it is relevant to explore whether known clinical prognostic factors can be combined with objective markers of sleep and circadian rhythms to predict the time to episode recurrence in BD. 

The study objectives were:(1)*Primary analyses: Time to mood recurrence.*(1a)To utilize a survival analysis to examine whether dimensions of objective estimates of sleep quantity, sleep variability and/or circadian rhythmicity (obtained using principal component analysis of actigraphy estimates recorded during euthymia) could predict the time to new episode onset after adjusting for known clinical predictors of BD outcome;(1b)To determine which, if any, individual actigraphic parameters (derived from the sleep and/or circadian dimensions identified in the first survival analysis) were significantly associated with new episode onset.(2)*Secondary analysis: Early mood recurrence.*

To use receiver operating curve (ROC) analyses to examine whether individual parameters (identified in the primary analyses) predicted an increased likelihood of early recurrence.

## 2. Materials and Methods

This study was part of a larger research protocol entitled GAN (Genetics, Actimetry, and Neuropsychology in Bipolar Disorders). The protocol is registered on the ClinicalTrials.org (accessed on 10 December 2013) website (NCT02627404) and all procedures were approved by the French ethics committee (Comité de Protection des Personnes–Ile de France (IDRCB2008_AO1465_50 VI–Pitié Salpêtrière 118–08)) and the Commission Nationale de l’Informatique et des Libertés (this committee is responsible for the protection of personal data of research participants). 

### 2.1. Sample

Participants were recruited from a university-affiliated psychiatric department in Paris, France, between 2013 and 2018. Inclusion criteria were:−Age ≥ 18 years, −DSM-IV diagnosis of BD type I or II according to the SCID (Structured Clinical Interview for DSM Disorders) [26];−Euthymic for ≥3 months, with a current MADRS score < 8 (Montgomery and Asberg Depression Rating Scale) [27] and a current YMRS score < 8 (Young Mania Rating Scale) [28];−Willing and able to give written informed consent;−Currently treated with at least one mood stabilizer (lithium or anticonvulsants or atypical antipsychotics). −Exclusion criteria were:−Unable to undertake actigraphy monitoring for 21 consecutive days;−Current alcohol or substance misuse problems (excluding current tobacco use);−Inpatient treatment and/or any modification of mood stabilizer regime in the 3 months prior to the assessment;−Current employment involves nightwork or shiftwork; −Recent trans-meridian travel; −Currently diagnosed with a comorbid neurological or sleep disorder (narcolepsy, obstructive sleep apnea, restless leg syndrome) and/or being prescribed a non-psychotropic medication that can alter sleep and circadian rhythms (such as cortisone);−Any other mental or physical health problem that contradicted participation in the study.

### 2.2. Clinical Data

Using data collected from the SCID [26], symptom rating scales, and written and electronic case records, we extracted key socio-demographic and clinical variables that were potentially associated with recurrence plus variables that may confound objective sleep monitoring. The included variables were: age, sex, BD subtype, current mood symptoms (MADRS, YMRS), age at onset of BD, lifetime number of BD episodes, body mass index (BMI in kg/m^2^), risk of obstructive sleep apnea (as assessed using the Berlin questionnaire) [29] and number and class of mood stabilizers prescribed at the time of recruitment (categorized as: lithium, anticonvulsants and atypical antipsychotics). 

### 2.3. Actigraphy Recording

Study participants wore an actiwatch (the CamNtech AW-7) continuously on the non-dominant wrist for 21 consecutive days. An actiwatch is a device that resembles a wristwatch and contains an accelerometer that detects, scores and stores information about the intensity and timing of wrist movements over consecutive 24 h intervals. For this study, an epoch of 1 min and an “average” sensitivity threshold was chosen. Participants were shown how to press a button on the device when they went to bed at night and when they got up in the morning (as this enabled the investigators to determine rest periods). At the end of the period of recording, we used the Actiwatch Sleep and Activity software program (V7.28) to estimate:(a)*Sleep quality*, as measured using the total sleep time (TST), sleep onset latency (SOL), sleep efficiency (SE), fragmentation index (FI) and time spent awake after sleep onset (WASO). We estimated the mean values for each sleep parameter.(b)*Sleep variability*, calculated as the within-individual variability in each of the sleep quantity parameters (using the standard deviation of each estimate over the recording period).(c)*Circadian rhythmicity* was represented by the inter-day stability (IS; scores range from 0, indicative of a total lack of rhythm, to 1, indicative of perfectly stable rhythm), intra-day variability (IV; a measure of fragmentation of activity, with scores ranging from 0–2), L5 (average level of activity over the least active 5 h), M10 (average level of activity over the most active 10 h), L5 onset and M10 onset (onset of the least active 5 h and the most active 10 h), amplitude (Amp; the difference between M10 and L5 activity) and relative amplitude (RA; the difference between M10 and L5 activity divided by the sum of L5 and M10, with the reported ratio ranging from 0–1).

### 2.4. Follow-Up

The first recurrence was defined as the new onset of an episode of BD of any polarity (i.e., manic, hypomanic, mixed or depressive) that met DSM-IV criteria in terms of the number of symptoms, duration and functional impairment and that occurred in the interval between the study recruitment and the end of follow-up. A recurrence can have led to hospitalization or ambulatory care (daycare hospital and/or clinical appointments with psychiatrists). Data about recurrent BD episodes were collected using two complementary sources of information. The first and main source of information was a computerized medical file in which psychiatrists systematically collected information about any mood recurrence that fulfilled the above-mentioned criteria and that occurred since the last visit. The second source of information was the hand-written case notes taken at each visit by the psychiatrists who were following each patient at the center. The identification of any recurrence was made by consensus between these two sources of information and two psychiatrists (LF and BE) solved any discrepancies. To minimize the attrition rate, each participant was called via phone by a nurse twice a year in order to organize a follow-up visit (if they did not already have more regular planned visits at the center with one of the psychiatrists). However, the final decision to attend the visit or not was the decision of the participant. The objective was to organize at least bi-annual follow-up visits for the longest follow-up period.

### 2.5. Statistical Analysis

Statistical analyses were performed using SPSS (version 26) and significance was set at *p* < 0.05. Sample characteristics were described using means (with standard deviations), medians (with interquartile ranges (IQRs)) or counts and percentages. Prior to undertaking the primary analyses, we examined the dataset for missing values. Less than 5% of individuals had any missing items and we determined that these data were missing at random. As such, missing values were replaced by sample means if necessary.

We first used a principal component analysis (PCA) to identify the sleep and circadian rhythms dimensions from the actigraphy recordings. This strategy was employed for two reasons. First, we previously demonstrated that different dimensions of sleep and circadian rhythms may discriminate BD cases from healthy controls [30]. Second, we were mindful of the need to ensure an acceptable “subject per variable” (SPV) ratio for the primary survival analysis (our SPV target was at least 1:5). As there were 18 actigraphic estimates, the use of PCA enabled us to identify key sleep and circadian rhythms dimensions and to employ a valid method of data reduction.

The next step involved selecting variables that may be predictors of mood recurrences: age, sex, BD subtype, residual depressive symptoms (with the MADRS score being log-transformed), current tobacco use, the density of BD episodes (number of episodes/duration of BD, where duration was estimated as current age minus age at onset, with the value being log-transformed) and number of mood stabilizers (MS monotherapy vs. MS polytherapy). In addition, we retained BMI and risk of obstructive sleep apnea (OSA) for the survival analyses (as these are known confounders of actigraphy estimates). We performed survival analyses with data censored at 60 months after the commencement of the study. We used a multivariate Cox proportional hazards model to estimate the adjusted hazard ratios (HRs) and 95% confidence intervals (CIs) for variables associated with time to recurrence. The model was constructed as followed: age and BD subtype were introduced into the model at the first step, then clinical covariates were entered at the second step with a backward stepwise option (BSLR) and, finally, the actigraphy dimensions (identified by PCA) were entered at the third step with a BSLR. Lastly, it was planned that, if any actigraphy dimension was a significant contributor to the final model, we would then re-run the model to look at the individual actigraphy metrics contained within that dimension. Each Cox proportional hazards model was then performed only with participants who had a recurrence during the follow-up (i.e., not taking into account “non-recurrent participants” who had a truncated duration of follow-up). This represented a “complete-case approach” that was the simplest, however, with a reduced sample size. This approach was used as a sensitivity analysis for comparison with the main analyses.

Planned secondary analyses examined the overall accuracy of the clinical and sleep and circadian rhythms variables (identified via survival analyses) in predicting early vs. late recurrence. We categorized groups as early and late recurrence (according to whether new episode onsets occurred before or after the median survival time) and examined the proportion of individuals who were correctly classified using receiver operating curve (ROC) analysis. We reported the ROC analysis as the area under the curve (AUC) with a 95% CI for each different combination of variables.

## 3. Results

### 3.1. Descriptive Analyses of the Sample

As shown in Table 1, the sample comprised 69 individuals with BD. About two-thirds were female (n = 59.4%) and about four-fifths met the criteria for BD type I (n = 78.3%). The sample median age was 42 years (IQR: 34–55), whilst the median age at onset of BD was 23 years (IQR: 19–30). Given the study inclusion criteria, the severity of mood symptoms was very low with a median MADRS score of 2 (IQR: 0–3.5) and a median YMRS score of 1 (IQR: 0–2). The median BMI was 24.7 kg/m^2^ (IQR: 22.7–27.9). About one out of five individuals met the criteria for a high risk of OSA (21.7%). About half of the study sample was current smokers (44.9%). Fifty-six individuals were prescribed lithium (81%), 27 were prescribed anticonvulsants (39%) and 20 were prescribed atypical antipsychotics (29%); 28 individuals were receiving MS monotherapy (41%). As shown in Table 2, the median TST was 482 min with a median SOL of 12 min. The median WASO was around 50 min, with a SE of about 85%.

### 3.2. Principal Component Analyses of Actigraphy Variables

We undertook two PCAs using standardized values of actigraphy estimates to identify the dimensions of sleep quality (SQ) and circadian rhythms (CR) (see Appendix A for further details of the PCAs). Each PCA extracted three factors. The three SQ factors accounted for 80% of the explained variance. Similarly, the three CR factors accounted for 83% of the explained variance. For sleep quality, the three dimensions were labeled as: SQ1 (seven actigraphy parameters mostly related to sleep discontinuity, including means and SDs), SQ2 (SOL mean and SOL SD) and SQ3 (mainly TST). For circadian rhythms, the three dimensions were labeled as: CR1 (which contained IV, IS, M10 and amplitude), CR2 (L5onset and M10onset) and CR3 (L5 and relative amplitude).

### 3.3. Survival Analyses

The median duration of follow-up was 3.5 years (IQR: 2–4.5). The attrition rates were 11.6% after year 1, 23.2% after year 2, 42% after year 3 and 63.8% after year 4 (see Appendix A for details). As shown in Figure 1, almost two-thirds of participants experienced a recurrent BD episode during follow-up (n = 44, 63.8%), with 24 individuals experiencing a major depressive episode and 20 experiencing a hypo/manic episode. The median time to first recurrence was 17.8 months (IQR: 6.2–60.7).

As shown in Table 3, the Cox survival analysis identified four predictors of recurrence: MS polytherapy (adjusted HR = 2.27, 95% CI: 1.18–4.38), BMI (adjusted HR = 0.86, 95% CI: 0.78–0.95), density of mood episodes (adjusted HR = 5.80, 95% CI: 1.39–24.28) and CR1 (adjusted HR = 0.66, 95% CI: 0.49–0.90).

When actigraphy estimates from the CR1 dimension (IV, IS, M10 and amplitude) were re-entered into the Cox model, it was found that IV was the key variable in the CR1 dimension (adjusted HR = 4.65, 95% CI: 1.39–15.51, *p* = 0.012) (see Appendix A for details).

Some participants were lost at follow-up (i.e., right-censored data). We performed the same Cox proportional hazards models only among the 44 participants who had a recurrence during the follow-up. Similar results were observed, i.e., associations between time to recurrence and CR1 (adjusted HR = 0.45, 95% CI: 0.29–0.68, *p* < 0.001) and between time to recurrence and IV (adjusted HR = 93.69, 95% CI: 12.1–680.16, *p* < 0.001).

### 3.4. ROC Analyses of Early and Later Recurrence

To generate the ROC analyses, we excluded eight participants with no recurrence before the median duration of 18 months. As shown in Figure 2 and Appendix A, we found that clinical variables alone were relatively poor at identifying individuals at risk of early vs. late recurrence (AUC = 0.64, 95% CI: 0.50–0.78). In contrast, IV alone correctly classified 75% of participants with early vs. late recurrence (AUC = 0.75, 95% CI: 0.62–0.88) and the classification was marginally improved when IV was combined with key clinical variables (AUC = 0.82, 95% CI: 0.72–0.92). Statistical analyses of the differences between the AUC demonstrated that the incremental improvement in correct classifications achieved with the IV plus clinical variables model (as compared with clinical variables alone) was of borderline significance (*p* = 0.04).

## 4. Discussion

This study investigated the associations between actigraphy estimates of sleep quality, sleep variability and circadian rhythms in a sample of 69 individuals with BD who were euthymic at baseline assessment and time to a mood episode recurrence during a median follow-up period of 3.5 years. We identified that an actigraphy dimension that comprised estimates of variability, stability and amplitude of circadian rhythms was significantly associated with the time to first recurrence. Among these estimates, a greater IV (intra-day variability) was the key parameter associated with time to recurrence. Furthermore, earlier onset of recurrence was predicted by greater IV alone (AUC = 0.75) or by IV in combination with other clinical variables (age, BD subtype, mood stabilizer polytherapy, density of mood episodes and BMI; AUC = 0.82).

The current research is important for three reasons. First, there are very few studies of actigraphy in BD that explored longitudinal outcomes. Therefore, this issue is an important gap in the knowledge base that warrants further research [19,22,31,32]. Second, our finding that the robustness of circadian rhythms may be significantly associated with time to recurrence overlaps with findings reported in a review of 27 studies that included data about longitudinal outcomes in individuals participating in actigraphy [22] and supports the findings of a recent larger-scale independent study [23]. Third, the current research builds on the methodology and findings of this latter study in several ways [23]. For example, we examined sleep and circadian dimensions instead of individual markers, we controlled for putative confounding factors (i.e., those likely to influence both actigraphy estimates and risk of recurrence) and we explored which combinations of actigraphy and clinical variables might more accurately predict time to recurrence.

Our findings can be viewed in the context of the wider literature on rest–activity rhythms in individuals with BD. For example, genetic studies have characterized circadian activity rhythm and sleep pattern phenotypes in individuals with BD and suggested that these phenotypes show familial loading [33] and may be trait characteristics of BD. Other large-scale community cohort studies demonstrated associations between the amplitude and stability of circadian rhythms in BD [34]. Furthermore, systematic reviews and meta-analyses of cross-sectional studies of individuals with BD compared with healthy controls repeatedly suggest that circadian dysrhythmias are more common in patients, even in euthymia [16,35]. Nevertheless, there is still some debate about the state or trait nature of these abnormalities. Our findings now add a new element to this literature. In our study, we found that individuals who met the criteria for euthymia (i.e., with very low levels of or no current BD symptoms), but who showed instability of and/or lower amplitude of circadian rhythms, had a shorter survival time to recurrence. The current symptom profiles of these patients would not necessarily have predicted recurrence and, thus, clinicians would have been reliant on the patient’s history (more frequent recurrences, treatment response, etc.) to try to identify the risk of early recurrence. This study demonstrated that an objective measure of underlying rest–activity instability, even in the absence of overt clinically observed mood instability, may assist in identifying those individuals who will experience an earlier recurrence. Of course, the sample size dictated that we must consider this hypothesis as interesting rather than proven and we acknowledge that bidirectional associations may be discussed between, e.g., the density of prior episodes and circadian disturbances. However, this is an important new avenue for future studies.

Considering the above, we also note that the ROC analyses suggested for the first time that, whilst clinical variables alone were rather imprecise predictors of BD outcome (a finding in agreement with [5,7]), the combination of these clinical variables with an objective marker (IV) derived from actigraphy recordings had a sufficient level of accuracy (>80%) to meet accepted criteria for use of a “diagnostic test” in clinical practice [36]. If these findings are replicated in larger samples, these actigraphy markers (and the use of dimensions) might provide a relatively simple way to identify individuals at higher risk of recurrence in clinical settings, and/or assist in the development of an individual risk predictor algorithm.

Of note, IV is an actigraphy estimate that quantifies the intra-daily rhythm fragmentation, i.e., the frequency and extent of transitions between periods of rest and activity on an hourly basis. High IV values may indicate, for example, the occurrence of daytime naps and/or nocturnal activity episodes. Healthy controls would typically have a unique prolonged period of activity during the day and a unique prolonged period of inactivity during the night. Inversely, individuals with BD may present inactive periods during the day and fragmented or agitated sleep during the night, which both contribute to higher values of IV. For instance, data using subjective or objective assessments of activity (self-report questionnaire or actigraphy) suggested that individuals with BD were the most physically active; however, they spent more time being sedentary [37] as compared to individuals with schizophrenia or major depression. Moreover, individuals with BD also showed high activity patterns in the morning and low activity patterns in the evening [38]. These observations translate into greater intra-daily rhythm fragmentation and increased values of IV.

Based on their circadian profile, “at-risk” individuals might then be oriented to more intensive monitoring and/or chronobiological interventions that would target some aspects of circadian rhythms [37]. Given our results of the association between “robustness” of circadian rhythms and time to recurrence, we hypothesize that any psychosocial intervention that encourages physical activity (to increase M10 and amplitude), reduces napping, prolongs inactive periods during the day and/or nocturnal activity (to decrease IV), and promotes regular activities and routines from one day to another (to increase IS) would in theory help to improve the outcome. All these actions are typical components of manualized psychoeducation therapy or interpersonal and social rhythms therapy, but should also be part of the typical recommendations made by psychiatrists to patients in clinical routine practice. For instance, group psychoeducation may have positive effects on biological rhythms (sleep/social domain) [39]. Interpersonal and social rhythm therapy that more directly targets social rhythms may increase the stability of such rhythms, which is thought to be important in reducing the recurrence of manic and depressive episodes [40]. However, clinicians will need support with deciding which interventions to employ and it is noticeable that a recent review of clinical practice guidelines for BD indicates that most of them fail to adequately address the management of problems related to sleep, circadian rhythms, activity and energy, and healthy lifestyles [38]. Finally, the fact that we identified several actigraphy estimates as predictive of the time to recurrence does not necessarily imply that any modification of these parameters would decrease the risk of recurrence. This should be tested in future clinical trials.

Whilst several psychological interventions that target BD are purported to stabilize circadian rhythms, the actigraphy variables we identified might also be targeted by medications. For instance, several reviews suggested that lithium (in humans and in animals) may influence the amplitude, stability and timing of circadian rhythms, although few studies focused on BD only [41,42]. However, small-scale studies undertaken over two decades ago suggested that the prophylactic effect of lithium may derive from slowing or delaying an overly fast circadian clock to prevent further desynchronization [43]. More recent studies suggested that lithium, but also quetiapine, affects several circadian parameters, including the robustness of circadian rhythms and peak activity time [44]. Moreover, cross-sectional analyses of data about individuals who demonstrate a good response to lithium suggested that they have more stable or robust circadian rhythms as compared to partial or non-responders [45,46]. We may, therefore, hypothesize that some individuals with more robust circadian rhythms would be less at risk of recurrences (i.e., lithium responders) because lithium had reversed a preexisting “at-risk” circadian profile. However, the cross-sectional design of these two later studies, but also the partial overlapping of used samples with the one from this study, lead us to stay very cautious about interpretations.

This study had several strengths, notably the long duration of follow-up (sufficient to capture new episodes in most of the sample) and 21 days of actigraphy recording (that allowed for reliable and accurate estimates of the variability of rhythms). Moreover, we evaluated several clinical and socio-demographic variables that may influence both actigraphy estimates and the risk of recurrence. Nevertheless, several limitations should be discussed. First, the sample size of this study did not exclude the possibility of some false negative statistical findings, which is relevant in the context of two sleep and circadian dimensions (SQ2 and CR2) that were of borderline significance in the multivariable model. In particular, we did not find any association between sleep estimates and time to recurrence (i.e., borderline *p*-value obtained with SQ2). This lack of association between the actigraphy estimates of sleep and time to recurrence was nevertheless consistent with results from a previous study [23]; however, it was not consistent with the few studies that suggested associations between subjective sleep measures and time to recurrence. These discrepancies between objective and subjective sleep measures as predicting (or not) the outcome would deserve future investigations. Second, whilst nearly two-thirds of participants experienced a recurrence during follow-up, we did not undertake separate survival analyses of predictors of recurrence according to polarity. Indeed, we did not plan to do this as we were aware (from the outset) that the sample size would not be sufficient for such an approach. It means that we cannot comment on whether any objective predictors might differentiate manic from depressive recurrences. Of note, a previous study [23] suggested that amplitude and M10 were associated with the time to depressive recurrence, while IV was associated with the time to (hypo)manic recurrence. Third, although we recorded prescriptions for MS (and included this in the analyses), we did not have details of adherence, dose, duration or changes in regime over time. Fourth, we used PCA and SPV ratios to enable us to rationalize the number of variables included in the multivariable analyses, but we did not apply any additional statistical corrections for multiple testing.

## 5. Conclusions

This study used actigraphy to identify dimensions of sleep and circadian rhythms that were associated with the longitudinal course of BD. We found that a circadian dimension (estimates of variability, stability and daily amplitude of circadian rhythms) was associated with time to first recurrence during the follow-up and that a particular variable (i.e., greater IV) accurately predicted earlier recurrence when combined with a few selected clinical variables. These findings and those from one recent publication [23] highlighted that actigraphy could be used in both clinical and research settings to increase the accuracy of the prediction of the outcome beyond typical clinical predictors.

## Figures and Tables

**Figure 1 jcm-11-02204-f001:**
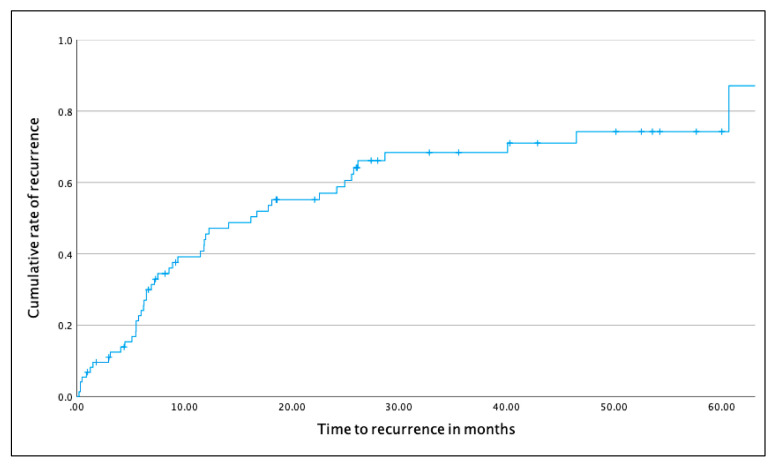
Kaplan–Meier survival curve showing the cumulative rate of recurrence over time (data censored at 60 months).

**Figure 2 jcm-11-02204-f002:**
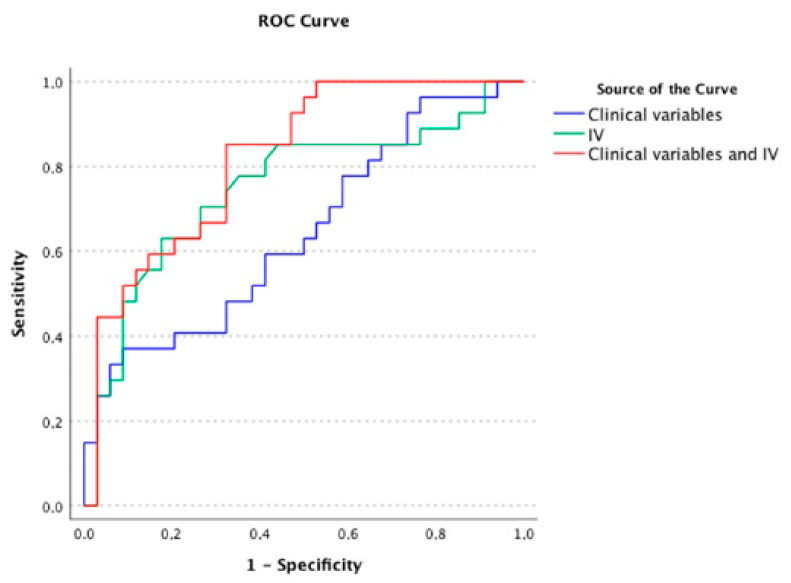
AUC of early versus later recurrence when classified using key clinical and circadian variables alone and in combination. Clinical variables: age, type of BD, density of mood episodes, mood stabilizers monotherapy and body mass index; IV: intra-daily variability.

**Table 1 jcm-11-02204-t001:** Socio-demographic and clinical characteristics of the sample (N = 69).

Variables	N	%	Median	IQR
Females	41	59.4		
Current age			42	34–55
BD type (type 1)	54	78.3		
Density of mood episodes (N/year)			0.52	0.3–0.9
MADRS			2	0–3.5
YMRS			1	0–2
Current tobacco use	31	44.9		
High risk of OSA	15	21.7		
BMI (kg/m^2^)			24.7	22.7–27.9
Mood stabilizer monotherapy *	28	40.6		

IQR: interquartile range, BD: bipolar disorder, MADRS: Montgomery–Asberg Depression Rating Scale, YMRS: Young Mania Rating Scale, OSA: obstructive sleep apnea, BMI: body mass index, N: number. * Mood stabilizer: lithium or anticonvulsants or atypical antipsychotics.

**Table 2 jcm-11-02204-t002:** Sleep and circadian rhythms characteristics of the sample.

Variables	Median	IQR
**Sleep variables**		
TST (min)	482	444–525
WASO (min)	52	33–66
SOL (min)	12	8–19
SE (%)	85	82–88
FI	30	24–37
**Sleep variables’ variability**		
SD TST (min)	87	67–108
SD WASO (min)	19.32	12.91–28.35
SD SOL (min)	14.77	8.57–28.15
SD SE	6.24	3.47–9.51
SD FI	9.19	7.58–13.11
**Circadian rhythms**		
IS (range: 0–1)	0.46	0.37–0.54
IV (range: 0–2)	0.83	0.69–0.92
M10 onset (h: min)	9:00	8–11
L5 onset (h: min)	1:00	0–2
M10	15,132	12,195–20,319
L5	890	582–1343
Amplitude	14,085	11,207–18,939
Relative amplitude (range: 0–1)	0.89	0.84–0.93

TST: total sleep time, SOL: sleep onset latency, WASO: time spent awake after sleep onset, SE: sleep efficiency, FI: fragmentation index, SD: standard deviation, IS: inter-day stability, IV: intra-day variability.

**Table 3 jcm-11-02204-t003:** Multivariable survival analysis (Cox regression model) using actigraphy factors.

Variables	Beta	SE	Wald	df	*p*	HR **	Lower 95% CI	Upper 95% CI
Age	0.027	0.016	2.904	1	0.09	1.03	0.99	1.06
Type BD	−0.189	0.421	0.201	1	0.65	0.83	0.36	1.89
**MS polytherapy**	0.821	0.335	6.020	1	**0.014**	2.27	1.18	4.38
**Density mood episodes ***	1.759	0.731	5.801	1	**0.016**	5.80	1.39	24.28
**BMI**	−0.151	0.053	7.938	1	**0.005**	0.86	0.78	0.95
SQ2	−0.381	0.201	3.597	1	0.06	0.68	0.46	1.01
**CR1**	−0.412	0.158	6.804	1	**0.009**	0.66	0.49	0.90
CR2	−0.353	0.203	3.036	1	0.08	0.70	0.47	1.04

*: log-transformed; ** adjusted HR; in bold: *p*-values < 0.05. SE: standard error, HR: hazard ratio, CI: confidence interval, BD: bipolar disorder, BMI: body mass index, SQ: sleep quality, CR: circadian rhythms, MS: mood stabilizers.

## Data Availability

Due to ethical and legal restrictions, data involving clinical participants cannot be made publicly available. All relevant data are available upon request to the authors for researchers who meet the criteria for access to confidential data.

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
