# Peer review of "Which Actigraphy Dimensions Predict Longitudinal Outcomes in Bipolar Disorders?"

_jcm, 2022, doi:10.3390/jcm11082204_

Round 1

Reviewer 1 Report

Interesting approach into the bipolar disease and in depth of other methods to predict recurrence in these patients.  

I suggest line 79 instead of repeating nevertheless exchange it for nonetheless to add some variety. 

Line 105 I suggest instead of To use survival analysis to examine to put:  Utilize a survival analysis to examine...

It will be more understandable to people that are not aware of what an actigraphy device is to make it more clear in the manuscript. Overall the manuscript is very interesting and shows that other methods can be applied to predict patterns and behaviors on bipolar patients. Best of luck!

Author Response

We thank the reviewer for these comments. Please find below our answers.

Reviewer 1

English language and style are fine/minor spell check required.

Answer

The revised version has been checked for spelling and grammar errors. 

I suggest line 79 instead of repeating nevertheless exchange it for nonetheless to add some variety. 

Answer

This has been modified.

Line 105 I suggest instead of To use survival analysis to examine to put:  Utilize a survival analysis to examine...

Answer

This has been modified.

It will be more understandable to people that are not aware of what an actigraphy device is to make it more clear in the manuscript. 

Answer

In the method section, we have added the following sentence to briefly describe what an actiwatch is.

"An actiwatch is a device that resembles a wrist watch. It contains an accelerometer that detects, scores and stores information about the intensity and timing of wrist movements over consecutive 24-hour intervals."

Reviewer 2 Report

This is an excellent study using objective actigraphy measures of sleep/circadian rhythms in a relatively large (N=64) sample of participants with bipolar disorder to longitudinally predict episode recurrence in a 3.5 year follow-up.  The study found that amplitude and variability in rhythms in 3 weeks of actigraphy were significant predictors of recurrence, suggesting rhythms might be targeted in interventions to prevent recurrence.

The study has a number of important strengths, including large sample, rare longitudinal follow-up from actigraphy measures, objective rhythm assessments rather than self-report/observer ratings, three weeks of sampling, careful control for other episode predictors, actigraphy confounds and separation of residual symptoms (esp sleep disruption) from recurrence. The many actigraphy measures were also reduced using PCA to create domain/factor scores, reducing risk of Type I error.  The actigraphy and other measurement methods were solid.  I only have minor suggestions.

The definition of recurrence could be clarified. “Hospital notes” were reviewed to determine recurrence; does this mean recurrences required hospitalizations or could a recurrence be symptom exacerbation described in outpatient notes at a hospital clinic?  What is the definition of a “full threshold episode” (p.4)? Notes were also reviewed for a 5-year period but the abstract says the follow-up was 3.5 years.

There were many, many actigraphy measures of sleep quality and rhythms but only one was a useful predictor (IV). Can the authors comment on the implications of sleep quality not being a good predictor and consistency of this finding with the literature?  Also, why only the one IV measure of rhythms?

Intra-day variability was the best (only?) predictor of recurrence.  Can the authors comment on why this particular variable is be best measured using actigraphy? Could it be measured more conveniently using self-report or observer ratings or even a retrospective report for the past month?  It is not that clear to a non-actigraphy reader what this variable is.  The authors mention that this possible actigraphy marker of recurrence can be measured “in a relatively simple way,” but rolling out actigraphy measures in a clinic, with good patient adherence, and then getting devices back and extracting and processing the data does not seem simple for psychiatry clinics. 

Related to this, how could psychosocial or pharmacologic interventions change (specifically) IV and what is the mechanism by which stabilizing IV would reduce recurrence? This is a potentially important finding, so explaining the model a bit more with regard to how improving circadian stability can improve outcome would be helpful.

Author Response

We thank the reviewer for these comments. Please find below our answers.

English language and style are fine/minor spell check required.

Answer

The revised version has been checked for spelling and grammar errors. 

The definition of recurrence could be clarified. “Hospital notes” were reviewed to determine recurrence; does this mean recurrences required hospitalizations or could a recurrence be symptom exacerbation described in outpatient notes at a hospital clinic?  What is the definition of a “full threshold episode” (p.4)?

Answer

We apologize for the lack of clarity. Two complementary sources of information about recurrences are available at the inclusion centre, since all participants had a (at least bi-annually) follow-up there. The first and main source of information is a computerized medical file in which psychiatrists systematically collect information about any recurrence that has occurred since the last visit and that fulfilled DSM-IV criteria for a mood episode, whatever the polarity of the episode and whatever the need of a hospitalization (i.e. any recurrence requiring in- or out-patient care). The second source of information is the case notes hand-written at each visit by the psychiatrists who are following the patient at the centre. The identification of recurrence was made by consensus between the two sources of information and two psychiatrists (LF and BE) solved discrepancies if any. In this study, we have considered that a recurrence is  a new episode that fulfilled DSM-IV criteria in terms of symptoms, duration, and functional impairment (which we have labeled as 'full threshold episode') and hospitalization is not mandatory to consider a recurrence that can have been treated in ambulatory care.

We have revised corresponding section that is now:

"2.4. Follow-up: First recurrence was defined as the new onset of a episode of BD of any polarity (i.e., manic, hypomanic, mixed or depressive) that met DSM-IV criteria in terms of number of symptoms, duration, and functional impairment and that occurred in the 5-year interval between the inclusion in the study and the end of follow-up. This recurrence can have led to a hospitalization or to ambulatory care (day care hospital and/or clinical appointments with psychiatrists). Data about recurrent BD episodes were collected using two complementary sources of information that are both available at the inclusion centre because all participants had regular (at least bi-annually) follow-up visits at the centre. The first and main source of information is a computerized medical file in which psychiatrists systematically collect information about any mood recurrence that fulfills the above-mentioned criteria and that has occurred since the last visit (whatever the polarity and independently of the need for an hospitalization). The second source of information is the hand-written case notes taken at each visit by the psychiatrists who are following each patient at the centre. The identification of recurrence was made by consensus between these two sources of information and two psychiatrists (LF and BE) solved discrepancies if any."

Notes were also reviewed for a 5-year period but the abstract says the follow-up was 3.5 years.

Answer

We reviewed the two sources of information up to five years when available. As mentioned in the manuscript, the median duration of follow-up was 3.5 years (IQR: 2-4.5). We have a few patients with follow-up data up to 5 years. We made no correction since we reported the median duration of follow-up consistently into the abstract and into the main text.

There were many, many actigraphy measures of sleep quality and rhythms but only one was a useful predictor (IV). Can the authors comment on the implications of sleep quality not being a good predictor and consistency of this finding with the literature?  Also, why only the one IV measure of rhythms?

Answer

When considering the final model presented in the manuscript, IV was indeed the only variable associated to the time to recurrence, while the first Cox model identified a factor labelled CR1 in which loaded not only IV, but also three other parameters (IS, M10 and amplitude). First, our sample size was moderate. Therefore, this does not preclude any false negative findings. As presented in table 3, one sleep factor (SQ2) that mainly contains sleep latency estimates was 'borderline' significant (p=0.06). With a larger sample size, we might have reached the significance threshold. Second, there are a few studies exploring the associations between time to relapse/recurrence and sleep disturbances/quality in BD, most of them suggesting associations. However, these studies mainly used subjective assessments of sleep, but not actigraphy. The only available study that used actigraphy estimates of sleep (Esaki et al. 2021) found no association between the outcome and sleep duration nor sleep efficiency, which is consistent with the results of our study. This may suggest discrepancies between results obtained with objective markers of sleep (that may not predict the outcome) and subjective markers of sleep (that may predict outcome). And this discrepancy would definitively deserve further investigation.

We have added the following sentences to the limitation section.

"In particular, we did not find any association between sleep estimates and time to recurrence (i.e. borderline p value obtained with SQ2). This lack of association between actigraphy estimates of sleep and time to recurrence is consistent with results from a previous study (Esaki et al. 2021), however not consistent with a few studies that suggested associations between subjective sleep measures and time to recurrence. These discrepancies between objective and subjective sleep measures as predicting or not predicting the outcome would deserve future investigations."

Regarding the second part of this comment (IV as the only measure of rhythms considered as a predictor), we performed separated analyses of each of the CR1 factor components (IV, IS, Amplitude and M10), with similar adjustments. The association with IV alone (as reported in the manuscript) was significant (p=0.012) as well as the one with IS alone (p=0.024), but not the ones with M10 alone (p=0.14) or amplitude alone (p=0.13). These estimates were inter-related (which is illustrated by their loading into the same CR1 factor) and the stepwise retained only IV.  As a consequence, our study suggests that IV was the only predictor of recurrence. As stated above, this does not preclude that other potential relevant markers (SQ2 and CR2) would have been significant with a larger sample size. Moreover, our approach (using a PCA and then "unpacking" factor components) has one advantage which is the data reduction in a sample, but may have missed some other parameters if studied one by one.

Intra-day variability was the best (only?) predictor of recurrence.  Can the authors comment on why this particular variable is be best measured using actigraphy? Could it be measured more conveniently using self-report or observer ratings or even a retrospective report for the past month?  It is not that clear to a non-actigraphy reader what this variable is. The authors mention that this possible actigraphy marker of recurrence can be measured “in a relatively simple way,” but rolling out actigraphy measures in a clinic, with good patient adherence, and then getting devices back and extracting and processing the data does not seem simple for psychiatry clinics. 

Answer

IV is an actigraphy estimate which quantifies the intra-daily rhythm fragmentation, i.e. the frequency and extent of transitions between periods of rest and activity on an hourly basis. High IV values may indicate for example the occurrence of daytime naps and/or nocturnal activity episodes. For example, healthy controls are supposed to have a unique prolonged period of activity during the day and a unique prolonged period of inactivity during the night. Inversely, individuals with BD may present inactive periods during the day and fragmentated or agitated sleep during the night which both contribute to higher values of IV. We could consider that, for instance, using EMA (ecological momentary assessment) of self-reported level of activity during the day, or using a sleep agenda to quantify napping periods or nighttime awakenings would be surrogates for this estimate. The main disadvantages of such assessments are 1) a less fine-grained quantification of phenomena (in our study the actiwatch recorded activity level every minute), 2) a tracking that mainly focuses on daytime activity when considering EMA) and 3) potential recall biases (when considering self-reports or interviews).

In this context, we consider that actigraphy recording is a quite easy way to reliably obtain this estimate, even in the psychiatry clinics. An actiwatch can be bought at a reasonable price (between 500-1000$) and is very easy to use. Its use requires two visits separated by 3 or 4 weeks (depending on the planned duration of recording), requires minimal intervention for the patient (who is only asked to press on the actiwatch button at bedtime and bedrise) and is very close to a passive collection of activity data. There is no major complexity for processing the data since the software will use the uploaded data to automatically estimate the parameters.

We made no major changes in the revised version but we provide a description of how IV can be interpreted in the discussion section. The added section is the following:

Of note, IV is an actigraphy estimate which quantifies the intra-daily rhythm fragmentation, i.e. the frequency and extent of transitions between periods of rest and activity on an hourly basis. High IV values may indicate for example the occurrence of daytime naps and/or nocturnal activity episodes. Healthy controls would typically have a unique prolonged period of activity during the day and a unique prolonged period of inactivity during the night. Inversely, individuals with BD may present inactive periods during the day and fragmentated or agitated sleep during the night which both contribute to higher values of IV. For instance, data using subjective or objective assessments of activity (self-report questionnaire or actigraphy) suggested that individuals with BD were the most physically active, yet spent most time being sedentary (Vancampfort, Firth et al. 2017) as compared to individuals with schizophrenia or major depression. Moreover, individuals with BD also showed high activity pattern in the morning and low activity pattern in the evening (Tanaka, Kokubo et al. 2018). These observations translate into greater intra-daily rhythm fragmentation and increased values of IV.

Related to this, how could psychosocial or pharmacologic interventions change (specifically) IV and what is the mechanism by which stabilizing IV would reduce recurrence? This is a potentially important finding, so explaining the model a bit more with regard to how improving circadian stability can improve outcome would be helpful.

Answer

Given the available literature, we can propose some information about how interventions (medications or psychosocial interventions) may modify circadian rhythms, but we have no information available to date specifically about IV. We also have to keep in mind that, while we identify IV as a relevant predictor, the first model identified a broader factor (i.e. CR1) as associated to the time to recurrence. CR1 comprised estimates of 'robustness' of circadian rhythms (more stable, less variable, with greater daytime amplitude) and we would refer to this dimension in the discussion rather than specifically to IV.

First, a whole paragraph on medications is already included in the submitted version: " Whilst several psychological interventions that target BD are purported to stabilize circadian rhythms, the actigraphy variables we identified might also be targeted by medications. For instance, several reviews suggested that lithium (in humans and in animals) may influence the amplitude, stability and timing of circadian rhythms, although few studies focused only on BD [39,40]. However, small scale studies undertaken over two decades ago suggested that the prophylactic effect of lithium may derive from slowing or delaying an overfast circadian clock to prevent further desynchronization [41]. More recent studies suggested that lithium, but also quetiapine, affect several circadian parameters, including peak activity time and robustness of circadian rhythms [42]. Moreover, cross-sectional analyses of data about individuals who demonstrate a good response to lithium suggest that they have more stable or robust circadian rhythms as compared to partial or non-responders [43,44]. We may therefore hypothesize that some individuals with more robust circadian rhythms would be less at risk of recurrences (i.e., lithium responders) because lithium had reversed a preexisting at-risk circadian profile. However, the cross-sectional design of these two later studies, but also the partial overlapping of used samples with the one from this study, lead to stay very cautious about interpretations". We tend to think that we gave here the main arguments about how medications may favor "robust" circadian rhythms. We would prefer not to extend too much this section because of the risk of being too speculative about any potential direct effects of medications on circadian rhythms.

Nevertheless, we agree that we have probably too briefly described how psychosocial interventions may modify circadian profile and then reduce the risk of recurrence. We have revised this section, however not specifically focusing on IV. Given the association between the outcome and the circadian dimension containing IV, IS, M10 and amplitude, we can hypothesize that any psychosocial intervention that promotes physical activity (to increase M10 and amplitude), reduces napping during the day and reduces nocturnal activity (to decrease IV), and promotes regular activity and routines from one day to another (to increase IS) would in theory help in decreasing recurrence risk. All these actions are typical components of manualized psychoeducation therapy or Interpersonal and Social Rhythms Therapy, but should also be part of the typical recommendations made by psychiatrists to patients in clinical routine practice.

The revised section is the following:

" Based on their circadian profile, identified individuals via these tests might then be targeted for more intensive monitoring and/or chronobiological interventions that would target some aspects of dysregulated circadian rhythms [37]. Based on our results of the association between "robustness" of circadian rhythms and time to recurrence, we can hypothesize that any psychosocial intervention that promotes physical activity (to increase M10 and amplitude), reduces napping during the day and reduces nocturnal activity (to decrease IV), and promotes regular activity and routines from one day to another (to increase IS) would in theory help to improve the outcome. All these actions are typical components of manualized psychoeducation therapy or Interpersonal and Social Rhythms Therapy, but should also be part of the typical recommendations made by psychiatrists to patients in clinical routine practice. For instance, group psychoeducation may have positive effects on biological rhythms (sleep/social domain) (Cardoso et al. 2015). Interpersonal and Social Rhythm Therapy that more directly targets social rhythms may increase stability of such rhythms, which is thought to be important in reducing recurrence of manic and depressive episodes (Bouwkamp Et al. 2012). However, clinicians will need support in deciding which interventions to employ and it is noticeable that a recent review of clinical practice guidelines for BD indicates that most of them fail to adequately address the management of problems related to sleep, circadian rhythms, activity and energy, and healthy lifestyles [38]. Finally, the fact that we identified several actigraphy estimates as predictive of the time to recurrence does not necessarily implies that any modification of these parameters would decrease the risk of recurrence. This would deserve to be tested in future clinical trials."